# Consumption of Meat in Brazil: A Perspective on Social Inequalities and Food and Nutrition Security

**DOI:** 10.3390/ijerph21121625

**Published:** 2024-12-05

**Authors:** Samantha Marques Vasconcelos Bonfim, Marhya Júlia Silva Leite, Isabela Gonçalves Camusso, Dirce Maria Lobo Marchioni, Aline Martins Carvalho

**Affiliations:** Department of Nutrition, Faculty of Public Health, University of São Paulo (USP), São Paulo 01246-904, SP, Brazil; samantha.mbonfim@usp.br (S.M.V.B.); marhyajuliasilvaleite@hotmail.com (M.J.S.L.); isabela.camusso@usp.br (I.G.C.); marchioni@usp.br (D.M.L.M.)

**Keywords:** meat, food intake, diet, health, sustainable development indicators

## Abstract

The EAT–Lancet “Planetary Health Diet” (PHD) proposes dietary recommendations to address health and environmental concerns, including reducing meat consumption. However, in Brazil, where meat holds cultural significance, the feasibility of these recommendations is questionable. This study aimed to examine meat consumption across the five Brazilian regions through the lens of the PHD, considering regionalisms and social inequalities. Using data from the 2017–2018 Household Budget Survey (POF), we estimated meat consumption. A multiple logistic regression analysis was employed to assess the association between excessive meat consumption and sociodemographic factors, such as region of the country, sex, per capita income profile, and level of Food and Nutrition Security (FNS). Our results reveal that meat consumption exceeds recommendations in all Brazilian regions. To effectively promote healthier and more sustainable diets, public health interventions must consider regional disparities and the cultural significance of meat. Policies should prioritize food justice and address the underlying social and economic factors that drive meat consumption.

## 1. Introduction

Red meat holds significant symbolic and cultural value [1,2,3], especially in Brazil [4]. However, its high consumption has been associated with negative health impacts [5,6,7,8,9] and environmental issues [6,7,8,9,10]. Regarding health detriments, numerous authors indicate that excessive consumption of red meat is associated with the development of diseases, including stroke [8,9,11,12,13,14], type 2 diabetes [8,9,11,12,13,14,15,16], and cancer [5,8,9,11,12,13,14,16,17]. As for the environment, excessive red meat consumption contributes significantly to climate change due to the potent greenhouse gas (GHG) emissions released throughout its production [3,4,7,8,9,16,18,19]. 

Reducing red meat consumption in diets is necessary for nutritional adequacy and the effective reduction of greenhouse gas emissions in Brazil, which are harmful to the environment [3,20,21]. In this regard, poultry, fish and seafood, eggs, and legumes are valuable sources of protein and micronutrients, and are considered alternatives to red meat consumption [8,9]. In the case of fish and seafood, studies indicate that their intake is related to a reduced risk of cardiovascular diseases due to the presence of omega-3 fatty acids, which, as precursors of eicosanoids, perform an essential function and also regulate heart rate, type 2 diabetes, impaired cognitive function, and cancer [22].

According to the FAO, a healthy and sustainable diet promotes all dimensions of individual health and well-being, generates low environmental impact, is accessible, safe, and equitable, and is culturally acceptable [11]. The consumption of animal source protein foods, when incorporated into a healthy dietary pattern, can play a pivotal role in achieving the Sustainable Development Goals (SDGs) [23]. Based on this concept and reasoning, a group of 37 renowned scientists from 16 countries, representing diverse fields including human health, agriculture, political science, and environmental sustainability, was assembled to develop a comprehensive report setting forth universal scientific targets for healthy diets and sustainable food production that would be applicable to all people on the planet [24]. The resulting document, “Food in the Anthropocene: the EAT–Lancet Commission on healthy diets from sustainable food systems”, published in 2019, proposes the “Planetary Health Diet” (PHD), which considers both health and environmental aspects, including recommendations for various protein sources [9]. In general, this diet is characterized by a predominance of plants, such as fruits, whole grains, legumes, nuts, and unsaturated oils, along with moderate amounts of seafood and poultry, and minimal consumption of red meat, processed meat, added sugar, refined grains, and starchy vegetables [9]. 

According to Willet et al. (2019), although based on health considerations and consistent with many traditional dietary patterns, for some individuals or populations the PHD may appear to be extreme or unfeasible; the authors claim that these guidelines have high potential for local adaptation and scalability, but they recognize that the changes required for their establishment need to be carefully considered, taking into account each local context and regional realities [9]. This is especially true regarding the adoption of diets with reduced amounts of meat (or no meat) [3,9]. Brazil, for instance, is a country with continental dimensions, divided into 27 Federative Units (UFs), comprising 26 states and the Federal District, and grouped into five major geographic regions: North, Northeast, Midwest, Southeast, and South. This division into five major Brazilian regions was established based on shared characteristics, considering physical, human, economic, and cultural aspects [25]. 

The Northern region is predominantly composed of the Amazon biome, a forested domain [25], where indigenous peoples, particularly in agricultural villages, traditionally cultivate roots (yams, cassava) and engage in daily fishing [26]. The Northeast features the semi-arid region [25], characterized by a diet combining beans and flour or rice, flour porridge with fats, meats (including offal), and stewed or fried fish [26]. These two regions exhibit the highest prevalence of severe food insecurity (moderate or severe)—45.2% in the North and 38.4% in the Northeast [27]. The Midwest has received financial incentives for agricultural production since the 1960s, becoming Brazil’s “agribusiness capital”. Originally inhabited by peasants and indigenous peoples, it encompasses the Cerrado and Pantanal, biomes rich in biodiversity that are threatened by the expansion of agribusiness [28]. Typical dishes include rice with pequi, beans with sausage, and grilled fish [29]. The Southeast boasts one of the highest population densities [30] and economic activity levels in the country [25,30]. The South encompasses the Pampa biome, with its population density linked to its border location and European immigration [25], and a dietary preference for fried and grilled foods, such as churrasco, the region’s signature dish [26]. In these latter two regions, the food culture is a result of the blending of habits from various immigrant groups, including Germans, Italians, Japanese, Poles, Syrian–Lebanese, and Russians [26]. Regarding moderate and severe food insecurity, approximately 4 out of 10 families in the North and Northeast, 3 out of 10 in the Midwest and Southeast, and 2 out of 10 in the South reported a partial or severe reduction in food consumption in the three months prior to the II National Survey on Food Insecurity in the Context of the COVID-19 Pandemic in Brazil [27].

Brazilian cuisine has been historically characterized by its diversity and the influence of the distinct human and natural possibilities encompassed within the country [29,30]. Its typical preparations are associated with seasonings, which are not considered “food”, but rather culinary subsidies, representing regional inscriptions of the original daily diet [26,30]. The valuation of biodiversity (forest and agricultural), and the material culture of different peoples expressed in their cuisines and cultivation practices, as well as social movements and public policies aimed at preserving this wealth [31,32], are important for improving human diets and food security, and mitigating the effects of climate change [33].

Promoting FNS and achieving climate change mitigation targets requires joint actions from various sectors; for example, public policies that assist with various stages, from the production to the purchase and the consumption of healthier and more sustainable food [18,34]; the development of new technologies; changes in industry behavior; and changes in the eating behavior of individuals, with a reduction in meat consumption [18]. There are differences in meat consumption across multiple contexts of inequities and social vulnerabilities [1,2,3]. Within this, it is necessary to consider the multidimensionality of the concepts of food consumption and sustainability in the Brazilian context, in which meat has an important symbolic value related to status and sociability [2,4]. While the country faces persistent challenges of poverty and food insecurity [27,35], red meat consumption is predicted to increase over the next decade [19].

This study aims to examine and discuss meat consumption across the five Brazilian regions through the lens of the PHD, considering the intersectionality of the economic accessibility and cultural acceptability of recommendations, particularly in the context of regionalisms and social and economic disparities in Brazil.

## 2. Materials and Methods

This research was conducted based on the average consumption reported in two non-consecutive 24 h dietary recalls (R24h) from the 2017–2018 Household Budget Survey (POF), a nationwide study conducted by complex sampling, with the household and the individual as units of investigation. The microdata from POF 2017–2018 were obtained from the official website of the Brazilian Institute of Geography and Statistics (IBGE) [36]. The sample consisted of 46164 people over 10 years of age. The census sectors were grouped into household strata with geographic and socioeconomic homogeneity, and the number of sectors in each stratum was proportional to the number of households in the stratum. Home visits in each stratum were evenly distributed over the 12 months to account for seasonal variations in food consumption [37]. More information about the research and collection can be found on the IBGE website [36].

### 2.1. Diet Reference Value and Animal Protein Sources

The PHD base is 2500 kcal/day [9]. In this study, 2000 kcal/day was used as a reference, a value recommended by Normative Instruction No. 75/2020, which establishes the technical requirements for the nutritional labeling declarations of foods in Brazil [38].

The dependent variables comprised five categories of animal source protein foods: beef and lamb (including cattle, sheep, and goats), pork, poultry (chicken and other poultry), fish and seafood, and total meat (the sum of all the previous categories). Processed meats, such as dried meat, jerked beef, ham, bologna, and nuggets, were computed to their animal origin (e.g., nuggets were classified as poultry, bologna as pork). This decision to omit a distinct “processed meat” category stemmed from two factors: first, the EAT–Lancet recommendations do not individually address processed and ultra-processed foods; second, consumption of these foods within the Brazilian population remains relatively low, ranging from 11% to 20% of meat consumption [39]. To quantify consumption estimates, the gram quantities of these ingredients within the recipe database were tallied. These estimates were then standardized to grams per 2000 kcal per day to facilitate comparisons between groups. Therefore, the recommendations for food sources of protein of animal origin considered here were as follows (Table 1):

### 2.2. Meat Consumption and the PHD

For these five food categories, the average consumption amount, the median, and the percentage of people who consumed such foods were estimated. The results were compared with the PHD recommendation values. The recommended values for Beef and Lamb are equivalent to “beef and lamb” in the PHD; Pork to “pork”; Poultry to “chicken and other poultry”; Fish and Seafood to “fish”; and Total Meat is the sum of all of these previous categories (see Table 1).

### 2.3. Determinants of High Meat Consumption

Multiple logistic regressions were used to verify the factors associated with a higher chance of a high consumption (above the upper limit recommended by the PHD) of all the categories of protein source foods. The set of independent variables used in the analyses were: region of the country (North, Northeast, Southeast, South, and Midwest), sex (male and female), skin color or race (white and non-white (grouping of Black, Asian, Brown, Indigenous, and undeclared)), age group (child and adolescent (10 to 19 years), adult (20 to 59 years), and older adults (≥60 years)), area (rural and urban), per capita income profile (quartiles of per capita income), and level of Food and Nutrition Security (FNS—based on the classification of the Brazilian Food Insecurity Scale (EBIA) [40], which categorizes FNS into food security and three levels of food insecurity (mild, moderate, and severe) in the household.

Stata 17 software was used for the calculations, considering the complexity of the sample and the expansion factors, and using the survey command, applied in all calculations of mean, median, and multiple logistic regressions. A map was created with the average consumption (grams per 2000 kcal/day) of meat by region of Brazil, with the aid of QGIS 3.34.4 and Canva^(C) 2024^ 2.291.0 softwares.

For the multiple logistic regression analyses, a 95% confidence interval (CI) was adopted, with *p* < 0.05. The following parameters were used as references: the Southeast region; female sex; non-white race or skin color; the child and adolescent age group; urban areas; the first quartile of per capita income; and being in a situation of FNS.

## 3. Results

### 3.1. Consumption of Meat in Brazil Compared to the PHD Recommendation

Based on the average consumption reported in two 24 h recalls from the 2017–2018 Household Budget Survey (POF), 2% of the population did not consume any amount of total meat on the reported days. For total meat, over 50% of consumers exceeded the PHD’s maximum recommended value of 56.8 g per 2000 kcal/day. Also, more than 60% of people consumed more beef and lamb than recommended, while less than 10% exceeded the maximum recommended value for fish and seafood (Table 2).

### 3.2. Consumption of Meat by Region Compared to the Recommended PHD

Among the regions of Brazil, consumption was heterogeneous (Figure 1). The average consumption, in grams per 2000 kcal/day, of beef and lamb was highest in the Midwest region (107.3 g), followed by the North (92.0 g) and South (80.8 g) regions. The South and Midwest regions were the largest consumers of pork, with an average consumption of 41.2 g and 34.7 g, respectively. The North region was the largest consumer of fish and seafood (64.0 g/day). In all other regions, the average amounts of fish and seafood consumed were within the recommendation of the PHD.

### 3.3. High Consumption of Meat in Brazil and Its Associated Factors

Multiple logistic regressions were used to verify the chance of high consumption (above the possible range recommended by the PHD) of beef and lamb, pork, poultry, fish and seafood, and total meat, according to associated factors—region of the country, skin color or race, age group, area of residence, income profile, and level of FNS (Table 3).

Regarding beef and lamb consumption, individuals residing in the Midwest were twice as likely to exceed the upper limit recommended by the PHD (OR: 1.9, CI 95%: 1.6–2.2). This group was also more likely to be male, adult, and in the highest income quartile. Conversely, those living in rural areas and experiencing moderate or severe food insecurity were less likely to consume above the upper limit for beef and lamb. The skin color or race variable did not show a significant association with high consumption of beef and lamb.

With regard to pork consumption, individuals residing in the South region were 50% more likely (OR: 1.6, CI 95%: 1.4–1.8) to exhibit elevated consumption, as were males and those with an income higher than USD 176.99 (3rd quartile). Inversely, adults and older individuals were less likely to exceed the recommended upper limit for pork consumption compared to children and adolescents. The variable of race did not show a significant association with the amount of pork consumed.

Higher poultry consumption was observed among individuals living in the Northeast, South, and Midwest regions compared to the Southeast, and with an income of USD 97.64 to USD 176.98 (2nd quartile). No other analyzed characteristics demonstrated significant differences in poultry consumption above the recommended levels.

Consumption of fish and seafood above recommended levels was approximately five times higher (OR: 4.9, CI 95%: 4.0–6.0) among individuals residing in the North region. Adult or older age, rural residency, and severe food insecurity were also associated with a greater likelihood of excessive fish and seafood consumption. Sex and race did not significantly influence the amount of fish and seafood consumed.

Analysis of total meat consumption revealed that individuals living in the North region were twice as likely (OR: 2.3, CI 95%: 2.0–2.7) to exceed the upper limit deemed acceptable by the PHD compared to those in the Southeast. Being male, an adult or older adult, and residing in rural areas were also associated with higher chances of exceeding this limit. In contrast, people in moderate or severe food insecurity were less likely to consume more than the recommended upper limit of total meat compared to food secure people. Skin color and income profile did not significantly affect the amount of total meat consumed.

## 4. Discussion

On the days reported, 98% of Brazilians consumed some meat, with 50% exceeding the PHD’s tolerable maximum value recommendations. Beef and lamb were the most highly consumed meats, followed by poultry, pork, and, lastly, fish and seafood. Consumption varied considerably across the five regions of Brazil. For example, fish and seafood consumption was low, with the exception of the North region, which indicated a higher intake of this category, along with total meat.

The variations in eating behavior among the Brazilian regions are a reflection of the diversity of biological and sociocultural characteristics in the country [3]. Although there is a penetration of ultra-processed foods in all regions and social groups, there is resistance from a resilient cultural diversity, represented in the findings of this study and others, such as the high consumption of fish in the North region [3,41,42,43] and the higher consumption of beef and preparations based on this food in the Midwest [3,44,45,46].

It is relevant to further reflect on the relationship between fish and seafood consumption in the North region and levels of FNS. According to Jacob and colleagues (2023), the availability of these foods is seasonal, so that fluctuating environmental factors influence harvesting capacity. In areas where people depend on these food groups, periods of scarcity (which have been increasing due to climate change) can result in severe food insecurity. Thus, the loss of biodiversity and climate change generate environmental, health, cultural, and economic consequences for these populations, which necessarily divert part of their income to acquire basic food necessities in conventional markets [33].

Other sociocultural variables also impact excessive meat intake. For example, we noted that men presented a higher chance of excessive red meat (beef, lamb, and pork), and total meat intake than women. Historically, meat has been a symbol of masculine power, strength, and virility [47,48]. This association can be also observed in the difference in consumption frequency between men and women. Santin et al. (2022) used data from the National Health Research in Brazil (PNS), and indicated that 64% of women and 51% of men limited their red meat intake to three times a week. This disparity may be linked to the pressures of a patriarchal society and existing gender stereotypes in food consumption [46]. 

Other important and known variables associated with meat intake are income and FNS. We noted that a higher income increased the chance of eating only red meat. In the same vein, Ferreira et al. (2023), in a study also carried out in Brazil, assessed adherence to the PHD by applying the Planetary Health Dietary Index (PHDI) indicator [49], which contains 16 components and provides a score that varies from 0 to 150 points [49,50]. The results of this study indicated an association between red meat consumption and family income. The consumption of red meat was beyond that recommended by the EAT–Lancet Commission, resulting in low results for the PHDI indicator score, regardless of per capita family income categories and FNS level, revealing that the prevalence of red meat consumption could be explained by its strong connection to local food culture [50]. 

No associations were found between high meat consumption and skin color or race, which may be explained by the cultural significance of meat in the country. Furthermore, different types of meat at varying price ranges were included within the same meat category (i.e., in the case of beef, premium steak, shank, flank, and ribs were all categorized as beef, despite price variations among them). Both white and non-white people showed consumption levels exceeding the meat threshold considered as the maximum tolerable value recommended in the PHD (11.2 g/d of beef, 11.2 g/of pork, 46.4 g/d of poultry, 80.0 g/d of fish and seafood, and 148.8 g/d of total meat). However, the relationships between FNS levels and meat consumption were evident. Data from the II National Survey on Food and Nutrition Insecurity in the Context of the COVID-19 Pandemic in Brazil, conducted by the Brazilian Network for Research on Food Sovereignty and Nutrition [27], revealed that 65% of households headed by Black or Brown individuals experienced some level of FNS, while households headed by white individuals had an FNS rate of 46.8%. This demonstrates that skin color or race negatively impacts access to foods such as meat within the population.

Studies have revealed that citizens and consumers have little knowledge about many issues relevant to the sustainability of the meat production sector [51,52]. The consumption of healthy and sustainable diets presents great opportunities to reduce GHG emissions from food systems and improve health outcomes [10]. However, according to Hase-Ueta et al. (2023), the responsibility for and vulnerability to the challenges facing planetary health are unevenly distributed among countries. In Brazil, the relevant factors associated with meat consumption may be the restriction of purchasing power, food accessibility, rising prices, and income inequalities, influenced by local variables such as socioeconomic inequalities and the cultural meanings that food can assume [53].

The recent scenario of reduced meat consumption and increased food insecurity and hunger, resulting from the impoverishment of the Brazilian population [27]—concomitantly with the country’s position as the third largest producer of meat (beef, pork, and poultry) in the world in 2021 [54]—reflects the unsustainability of the predominant food system in Brazil. As Hase-Ueta et al. (2022) point out, it is relevant to investigate the reasons behind increases in meat prices over time, as well as to emphasize that this phenomenon is probably not the result of a food policy of meat reduction, nor of the internalization of environmental impact. In their study, the economic differences between high-income profiles and low- and middle-income profiles had an impact on what, how much, and why people ate [53].

In order for the population to adhere more closely to recommendations to reduce beef consumption and achieve climate impact mitigation targets, nutritional needs cannot be neglected, since the intensity of the environmental impact of food is just one of several criteria to be considered when making food choices [45,53]. The document “Contribution of terrestrial animal source food to healthy diets for improved nutrition and health outcomes—An evidence and policy overview on the state of knowledge and gaps” [23], and The Lancet Commission report titled “The Global Syndemic of Obesity, Undernutrition, and Climate Change” [7], recommend the consumption of animal source protein foods in moderate amounts as part of healthy and diversified diets in all phases of life, with adaptations based on context, considering cultural preferences, baseline nutritional status, and dietary patterns.

A strategy of charging the true costs of meat, in order to increase its price and reduce its consumption [7] in contexts like that of Brazil, would demonstrate only an environmental concern about the impacts of consuming this food. People with higher incomes could pay the difference in the price of meat, and a reduction in consumption could possibly be motivated by environmental awareness. Lower-income classes, on the other hand, would have their eating patterns changed due to an inability to choose what to eat. In this sense, the country’s political discourse is in favor of facilitating access to essential subsistence items with healthy and minimally processed foods for the population, so much so that, in 2024, meat in general was included as a component of the basic food basket [55]. This fact may cause concern for the environmental cause, but it should be celebrated as a step towards food security, as well as evaluated and discussed in terms of the social inequalities and inequities that exist in Brazil.

The transition to sustainable diets must encompass the debate on social and food justice as a high priority [53]. Beal et al. (2023) point out that animal source foods play relevant and distinct roles in achieving healthy and sustainable food systems in different contexts around the world. Thus, efforts are needed to ensure better production practices, reduce excessive consumption when it is high, and sustainably increase consumption when it is low [56].

Government agencies could implement fiscal interventions to achieve food sovereignty and nutrition security, such as promoting organic agriculture, reducing subsidies to companies that engage in deforestation, regulating food advertising, and improving food labeling and menus, with the potential addition of information on environmental impacts, as well as curbing the advertisement of products that are not consistent with dietary guideline recommendations [57,58,59] such as the 2014 Brazilian Dietary Guidelines [60], which present dietary recommendations focused on eating patterns that prioritize plant-based foods and have a reduced content of meat, sweets, sugars, and ultra-processed foods in general; however, there are no quantitative recommendations in the Brazilian guidelines [32,57,58].

It is evident that the FNS data, collected through the EBIA, constitute a limitation for this study, since they refer to the household level and not necessarily to an individual level, as prioritized in this work. However, the EBIA questionnaire allows us to assimilate how each degree of food insecurity can affect individuals in the five regions of the country and their possible relationships with meat. In addition, it is important to indicate the under-reporting of food consumption as a limitation—something that is common in population-based studies that use surveys such as 24 dietary recalls, and makes it impossible to predict which foods specifically will be subject to this bias. On the other hand, this is the most commonly used tool in population-based studies such as those of the POF, because it has a lower associated measurement error, in addition to a collection governed by strict methodological standards that aim to favor the quality of the information obtained [50]. In our study, neither race nor color demonstrated a significant association with excessive consumption of any type of meat. This may be attributed to the categorization of participants into only two racial groups (white and non-white) to achieve a representative non-white sample. Furthermore, the division of income into per capita income quartiles may not accurately reflect the realities of household income in Brazil. Different methods of income categorization may potentially produce divergent results.

Given the intricate nature of food choices, this study was unable to comprehensively assess all aspects and methods of meat consumption. Food consumption data alone are insufficient to evaluate factors such as affordability, intentionality, and individual choice. This study contributes to the literature by examining the influence of various sociodemographic characteristics on meat consumption in Brazil. Additionally, it serves as a comparative benchmark for future research on household consumption, planetary health, and the use of the POF dataset. Moreover, this study can inform the development of realistic and evidence-based dietary guidelines aimed at reducing meat consumption.

## 5. Conclusions

This study demonstrated that almost all Brazilians consumed some quantity of total meat, according the 24 h dietary recalls. Additionally, the average consumption of beef, lamb, poultry, and total meat exceeded the recommended tolerable intake of the PHD in all Brazilian regions. This finding raises concerns about potential negative repercussions for both public health and the environment, highlighting the need for interventions promoting healthy and sustainable dietary practices. Excessive consumption of fish and seafood was low, except in the North region, where elevated consumption was also associated with a greater likelihood of severe food insecurity.

The results also indicated that meat consumption is permeated by different social factors, such as geographic location, sex, income profile, and level of FNS. The chance of a high consumption of beef and lamb, pork, and total meat was greater among males. For beef and lamb, it was associated with the level of FNS—the amount consumed decreased as the degree of food insecurity increased.

Complex phenomena like excessive meat consumption require complex solutions. Considering that food choices are influenced by environment, culture, and opportunity, social policies and programs that directly or indirectly support the reduction of meat consumption are crucial for promoting adequate, healthy, and sustainable diets. However, these interventions will only be effective if they are tailored to the specific dietary patterns and nutritional challenges of a given region, guided by the principles of food justice. Only from localized identifications, through the co-production of knowledge with the participation of different sectors of society (rural producers, managers, politicians, entrepreneurs, and consumers, among other actors), can we understand the connections and relationships between the production of food and where and why it is consumed. Brazil is one of the largest producers of beef in the world, with a significant portion aimed at export. Therefore, even if the population reduces its red meat consumption, its effectiveness in terms of environmental issues has limits, which calls for interventions aimed at production systems with low environmental impacts.

## Figures and Tables

**Figure 1 ijerph-21-01625-f001:**
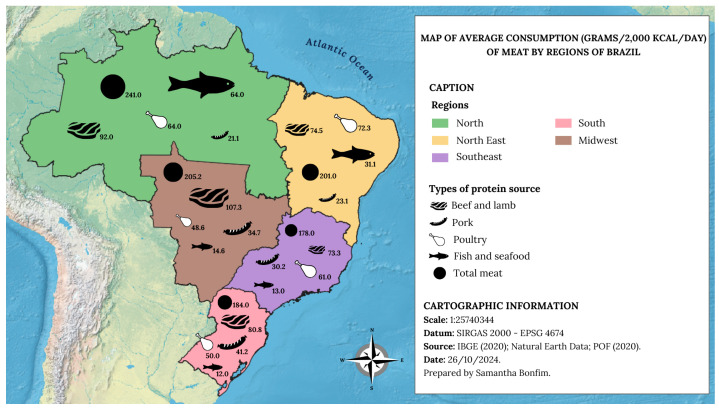
Map of consumption (g/2000 kcal/day) of meat by region of Brazil. Brazil, POF 2017–2018.

**Table 1 ijerph-21-01625-t001:** Recommended values for animal protein sources (g/day) from the “Planetary Health Diet”, with possible ranges, for an intake of 2000 kcal/day.

Sources of Animal Protein *	Consumption of Animal Protein Sources, Grams/Day (Possible Range)
Beef and lamb	5.6 (0–11.2)
Pork	5.6 (0–11.2)
Poultry	23.2 (0–46.4)
Fish and Seafood	22.4 (0–80.0)
Total meat	56.8 (0–148.8)

* Values based on a diet of 2000 kcal/day. Equivalences (study categories = EAT–Lancet category): Beef and Lamb = beef and lamb; Pork = pork; Poultry = chicken and other poultry; Fish and Seafood = fish; Total meat = sum of all previous categories. Source: Adapted from Willet et al., 2019.

**Table 2 ijerph-21-01625-t002:** Consumption of meat in Brazil compared to the “Planetary Health Diet” recommendation. Brazil, POF 2017–2018.

	Mean and Median Consumption in Grams *		Percentage of Consumption	Percentage of Consumption Above the “Planetary Health Diet” Recommendation (%)	Recommended Value of the “Planetary Health Diet” * (Possible Range)
Type of Protein Source	Mean	p(50)			
Beef and lamb	78.9	60.1	73.3	67.0	5.6 (0–11.2)
Pork	29.5	0.0	43.4	32.0	5.6 (0–11.2)
Poultry	61.7	0.0	57.7	43.5	23.2 (0–46.4)
Fish and seafood	22.0	0.0	17.8	9.4	22.4 (0–80.0)
Total meat	192.1	176.0	98.0	61.5	56.8 (0–148.8)

* Values based on a diet of 2000 kcal/day.

**Table 3 ijerph-21-01625-t003:** Consumption of meat in Brazil above that recommended by the “Planetary Health Diet” and associations with region, sex, skin color or race, age group, area of residence (rural or urban), income profile, and food security level. Brazil, POF 2017–2018.

	Beef and Lamb			Pork			Poultry			Fish and Seafood			Total Meat		
Characteristics	* OR	* CI 95%	*p*-Value *	OR	CI 95%	*p*-Value	OR	CI 95%	*p*-Value	OR	CI 95%	*p*-Value	OR	CI 95%	*p*-Value
Region															
Southeast	1.0	-	-	1.0	-	-	1.0	-	-	1.0	-	-	1.0	-	-
North	1.5	1.3–1.8	0.000	0.6	0.5–0.7	0.000	1.0	0.8–1.2	0.898	4.9	4.0–6.0	0.000	2.3	2.0–2.7	0.000
Northeast	1.2	1.0–1.3	0.009	0.7	0.6–0.7	0.000	1.2	1.1–1.4	0.000	2.1	1.8–2.5	0.000	1.3	1.1–1.4	0.000
South	1.4	1.2–1.5	0.000	1.6	1.4–1.8	0.000	0.7	0.7–0.8	0.000	0.8	0.6–1.1	0.131	1.1	1.0–1.2	0.206
Midwest	1.9	1.6–2.2	0.000	0.9	0.8–1.0	0.180	0.7	0.6–0.8	0.000	1.1	0.8–1.5	0.406	1.6	1.5–1.8	0.000
Sex															
Female	1.0	-	-	1.0	-	-	1.0	-	-	1.0	-	-	1.0	-	-
Male	1.2	1.1–1.2	0.000	1.2	1.2–1.3	0.000	1.0	1.0–1.1	0.911	0.9	0.8–1.0	0.059	1.2	1.1–1.3	0.000
Skin color or race															
Non-white *	1.0	-	-	1.0	-	-	1.0	-	-	1.0	-	-	1.0	-	-
White	1.1	1.0–1.2	0.069	1.0	0.9–1.0	0.430	0.9	0.9–1.0	0.087	1.0	0.9–1.1	0.830	1.0	0.9–1.0	0.400
Age group															
Child and adolescent	1.0	-	-	1.0	-	-	1.0	-	-	1.0	-	-	1.0	-	-
Adult	1.1	1.0–1.2	0.181	0.8	0.8–0.9	0.000	1.0	1.0–1.2	0.086	1.3	1.1–1.5	0.001	1.4	1.3–1.5	0.000
Older adults	0.8	0.7–0.9	0.002	0.5	0.5–0.6	0.000	0.9	0.8–1.0	0.174	1.4	1.1–1.7	0.001	1.1	1.0–1.3	0.050
Area															
Urban	1.0	-	-	1.0	-	-	1.0	-	-	1.0	-	-	1.0	-	-
Rural	0.8	0.8–0.9	0.002	1.1	1.0–1.2	0.068	0.9	0.8–1.0	0.066	1.4	1.2–1.6	0.000	1.3	1.2–1.5	0.000
Income profile (USD) *															
0.00–97.63	1.0	-	-	1.0	-	-	1.0	-	-	1.0	-	-	1.0	-	-
97.64–176.98	1.1	1.0–1.3	0.108	1.0	0.9–1.2	0.688	1.2	1.0–1.3	0.032	0.7	0.6–0.8	0.000	0.9	0.8–1.1	0.155
176.99–303.42	1.3	1.1–1.5	0.001	1.1	1.0–1.3	0.040	0.9	0.8–1.1	0.329	0.8	0.7–1.0	0.063	0.9	0.8–1.0	0.055
303.43–41,184.52	1.7	1.5–2.0	0.000	1.2	1.0–1.4	0.003	0.9	0.8–1.0	0.159	1.1	0.9–1.4	0.297	0.9	0.8–1.0	0.076
Food security level															
Food secure	1.0	-	-	1.0	-	-	1.0	-	-	1.0	-	-	1.0	-	-
Mild food insecurity	1.0	0.9–1.1	0.779	1.0	1.0–1.2	0.230	1.0	0.9–1.1	0.832	1.1	0.9–1.2	0.505	1.0	0.9–1.1	0.449
Moderate food insecurity	0.8	0.7–0.9	0.001	0.9	0.8–1.0	0.191	1.0	0.9–1.3	0.275	1.1	0.9–1.4	0.188	0.8	0.7–1.0	0.002
Severe food insecurity	0.5	0.4–0.6	0.000	0.8	0.6–0.9	0.002	0.9	0.8–1.0	0.225	1.7	1.3–2.2	0.000	0.8	0.7–1.0	0.009

* OR = odds ratio. CI 95% = 95% confidence interval. *p*-value < 0.05. Non-white = grouping of Black, Asian, Brown, Indigenous, and undeclared. Income profile = quartiles of per capita income. * Values based on a diet of 2000 kcal/day.

## Data Availability

The original data presented in the study are openly available on IBGE’s website at https://www.ibge.gov.br/pesquisa-de-orcamentos-familiares (last accessed on 5 November 2024).

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
