# Peer review of "Consumption of Meat in Brazil: A Perspective on Social Inequalities and Food and Nutrition Security"

_ijerph, 2024, doi:10.3390/ijerph21121625_

Round 1
Reviewer 1 Report
Comments and Suggestions for Authors
Using POF data, the article analyzes meat consumption in Brazil considering the recommendations of the EAT-Lancet "Planetary Health Diet". The text is original and addresses a topic of great relevance for the scientific field and for the development of public political strategies adapted to different sociocultural realities.
Some adjustments can improve the quality of the text for its publication.
Keywords: include “meat”.
Line 38: the word “which” appear twice.
It’s relevant to introduce in Introduction or methods a description/characterization of the different Brazilian regions, not only regarding socioeconomic aspects but also regarding food culture. Furthermore, it would be necessary to give more details about the EAT-Lancet "Planetary Health Diet". Why did the authors choose these recommendations? How important is it? Who and when prepared this document? Etc...
Line 216-220: the authors indicate “they found a positive relationship between the consumption of this food group and the male sex - in which the decision to acquire fish is impacted by family composition, mainly by the presence of children, adolescents, and households headed by men. In general, there were no major differences in household acquisition decisions between income brackets”. Historically, in different sociocultural contexts, the consumption of “white meat”, especially fish, has been associated with women and femininity. Why does this situation seem different in Brazil? Furthermore, in other cultural contexts, the economic situation also has an impact on fish consumption. How do you explain your results?
Line 227: better “economic” instead of “financial”.
Line 240: It would be useful to indicate that the study of Ferreira et al. (2023) was also carried out in Brazil.
Line 248-250: “Color and race were not associated with the consumption of any type of meat, which could be explained by the high degree of racial mixing in the Brazilian population”. This statement is problematic, because it reaffirms the “myth” of the harmonious miscegenation of Brazilian society. Even though there is racial mixing, the highest rates of poverty, unemployment, low level of education, etc. are still associated with black and brown populations. Furthermore, if there is a difference in red meat consumption in relation to income, how is this not also reflected in race?
Line 271-274: this sentence is difficult to understand.
Although the authors' aims were not to compare meat consumption with national recommendations, it would also be relevant to mention Brazilian recommendations on healthy eating and meat consumption, comparing them with the Eat Lancet recommendations. For example, how is this issue addressed in Brazilian dietary guidelines?
Conclusion:
The authors indicate: “The results also indicated that the consumption of meat is permeated by different social factors, inserted in a complex intersection of aspects such as geographic location, culture, gender, color or race, age, income, and level of food and nutrition security”. This complex intersection is not really presented in the article and culture is not deeply addressed.
Reviewer 2 Report
Comments and Suggestions for Authors
Dear author,
First, thank you for drawing attention to this topic, which is popular and whose importance is increasing.
The introduction of the full text and abstract can provide a more general introduction, and then detailed information can be given.
A more detailed sentence can be used to introduce the disadvantages of a meat-based diet.
The discussion section is very attentive and the gender-based comments are nicely integrated with other references and articles.
Comments on the Quality of English LanguageGenerally understandable. In some places, very long sentences can improve understanding negatively.
Reviewer 3 Report
Comments and Suggestions for Authors
Thank you for the opportunity to review the interesting manuscript titled Food Consumption of Meat in Brazil: A Perspective on Social Inequalities and Food and Nutrition Security.
The authors have addressed an interesting topic of excessive meat consumption in Brazil and investigated factors influencing meat consumption. The study was well designed, conducted, and it quite well described.
In the Abstract, it would be good to have examples of social factors that influence meat consumption.
Minor English revisions would improve the clarity and flow of the manuscript, e.g., not starting sentences with short conjunctions such as and, but, and or; unclear use of tense (l.66-67); and using empty fillers such as ‘it was found’ – just state the finding directly. I also recommend replacing the word ‘elderly’ that is often considered ageist with ‘older adults’, and use ‘skin color’ rather than ‘color’.
The manuscript provides important findings about the variation in the needs while considering sustainability and will contribute to our understanding of the facts and better targeting of future interventions, because the general recommendations may not fit all.
Comments on the Quality of English LanguageMinor English revisions would improve the clarity and flow of the manuscript, e.g., not starting sentences with short conjunctions such as and, but, and or; unclear use of tense (l. 66-67); and using empty fillers such as ‘it was found’ – just state the finding directly. I also recommend replacing the word ‘elderly’ that is often considered ageist with ‘older adults’, and use ‘skin color’ rather than ‘color’.
